# CNN-FWS: A Model for the Diagnosis of Normal and Abnormal ECG with Feature Adaptive

**DOI:** 10.3390/e24040471

**Published:** 2022-03-28

**Authors:** Junjiang Zhu, Jintao Lv, Dongdong Kong

**Affiliations:** 1School of Mechanical and Electrical Engineering, China Jiliang University, Hangzhou 310018, China; s20010811016@cjlu.edu.cn; 2School of Mechatronic Engineering and Automation, Shanghai University, Shanghai 200444, China; kodon180704@shu.edu.cn

**Keywords:** ECG diagnosis, convolutional neural network, feature extraction, feature adaptive screening

## Abstract

(1) Background and objective: Cardiovascular disease is one of the most common causes of death in today’s world. ECG is crucial in the early detection and prevention of cardiovascular disease. In this study, an improved deep learning method is proposed to diagnose abnormal and normal ECG accurately. (2) Methods: This paper proposes a CNN-FWS that combines three convolutional neural networks (CNN) and recursive feature elimination based on feature weights (FW-RFE), which diagnoses abnormal and normal ECG. F1 score and Recall are used to evaluate the performance. (3) Results: A total of 17,259 records were used in this study, which validated the diagnostic performance of CNN-FWS for normal and abnormal ECG signals in 12 leads. The experimental results show that the F1 score of CNN-FWS is 0.902, and the Recall of CNN-FWS is 0.889. (4) Conclusion: CNN-FWS absorbs the advantages of convolutional neural networks (CNN) to obtain three parts of different spatial information and enrich the learned features. CNN-FWS can select the most relevant features while eliminating unrelated and redundant features by FW-RFE, making the residual features more representative and effective. The method is an end-to-end modeling approach that enables an adaptive feature selection process without human intervention.

## 1. Introduction

The prevalence of cardiovascular disease in China continues to rise, and the number of people with cardiovascular disease is estimated to reach 330 million [1]. ECG testing is widely used due to its non-invasive and practical nature and is the preferred method of cardiovascular disease prevention. Although ECG sampling is currently easy, diagnosis using ECG usually requires a physician with expertise and personal experience. Considering the large number of people asking for an ECG, the workload for doctors is enormous. The effectiveness of physicians’ ECG diagnoses will be considerably improved if the majority of normal ECGs are filtered using computer-aided technology for clinical ECG detection.

The methods of computer-assisted ECG analysis have evolved rapidly in just a few decades. In the beginning, algorithms were based on the physician’s logical deduction, Such methods first extract the clinical features of the ECG, for example, R-wave duration [2], P-wave duration [3], T-wave duration [4], and other methods, and then these combined with medical rules to give conclusions. J.S. Geddes et al. [5] designed rules to achieve the classification of premature ventricular ECG. First, the parameters to be detected were selected according to the characteristics of ventricular premature ECG performance: R-R interval, QRS wave group duration, and QRS wave group shape. Then the following rules were used as criteria for judging the occurrence of ventricular premature beats: (1) when the QRS duration is slightly wider than the QRS duration standard, and the current R-R interval is judged to be <90% of the previous R-R interval, the diagnosis is ventricular premature beats; (2) if the QRS duration is significantly wider than the QRS duration standard, the diagnosis is ventricular premature beats; (3) when the QRS wave cluster has a different shape than the standard shape, the diagnosis is ventricular premature contraction. This method is more intuitive, but the extraction algorithm of clinical indicators other than r-wave is inaccurate and affects the conclusion.

Various mathematical and theoretical features have been explored to replace the obvious clinical indicators, and together with some nonlinear classifiers, a set of methods called pattern recognition has been developed to diagnose the ECG. These mathematical and theoretical features include, among others, Fourier transform [6], wavelet transform [7], Welch’s method [8], EMD [9], component analysis (PCA) [10], and independent component analysis (ICA) [11]. For example, R.J. Marti et al. [12] used Independent Component Analysis (ICA) for numerical feature extraction and PNN for arrhythmia diagnosis, with mean sensitivity, specificity, positive predictive value (PPV), and accuracy of 99.97%, 99.83%, 99.21%, and 99.28%, respectively, in the case of a tenfold cross-validation scheme. However, these method classes treat the extraction and classification sessions in isolation without considering their connection, and their accuracy often relies on the experience of the algorithm designers.

In recent years, deep learning techniques have overcome the challenges faced by traditional methods and have made a splash in automated ECG diagnosis because the methods themselves do not require the manual production of features for analysis, but rather the direct use of neural networks for classification. Many scholars have conducted in-depth research on ECG classification and diagnosis. Kiranyaz et al. [13] suggested that a one-dimensional CNN provides rapid and reliable patient-specific ECG classification and prediction. Acharya et al. [14,15] constructed CNN models to achieve ECG classification without conducting any feature selection. To accomplish ECG positive abnormality classification, Jin et al. [16] employed an ECG signal as an input signal and a CNN and R-wave feature extraction set as a diagnostic approach. To obtain successful ECG classification, Salloum and Kuo [17] used RNN for the first time in their study. Chen et al. [18] suggested a DNN-based method for classifying ECG positive abnormality signals. Khan et al. [19] employed principal component analysis (PCA) for denoising and LSTM for classification to classify arrhythmias. Oh et al. [20] suggested combining a convolutional neural network (CNN) with long short-term memory (LSTM) in a system to identify frequent arrhythmias on ECG with high accuracy. Hou et al. [21] suggested using LSTM-AE to extract ECG characteristics and SVM for ECG classification. Lee et al. [22] used the intrinsic mode function of SVM-RFE with 3 and 32 feature dimension methods for heartbeat recognition, showing robust performance. Hammad et al. [23] developed a new DNN technique based on cross-validation paired with GA for feature and parameter optimization to accomplish arrhythmia classification.

The above work demonstrates the usefulness of deep learning network structures in ECG diagnosis. However, in clinical practice, neural network models are prone to encounter problems such as inadequate representation of the features acquired by the deep learning network structure, poor adaptive filtering of features, and vague generalization of the classification model. This paper proposes a CNN-FWS that combines three convolutional neural networks (CNN) and recursive feature elimination based on feature weights (FW-RFE). CNN-FWS optimizes feature information representation and selects the most relevant features while eliminating unrelated and redundant features by implementing adaptive features filtering. ECG is processed into three stages: spatial information extraction using a variety of convolutional neural networks (CNNs), spatial information optimization using recursive feature weight elimination (FW-RFE), and diagnosis of fully connected layers. The PTB-XL ECG dataset [24] was used to verify the generalization performance of the model of this paper.

This paper is organized as follows. Section 2 introduces the related theoretical methods and the proposed integrated convolutional neural network (CNN) and recursive feature elimination based on feature weight (FW-RFE). Acquisition and processing of human ECG data are presented in Section 3. Section 4 presents the results. Section 5 provides a discussion based on the results obtained in Section 4. Finally, Section 6 concludes the paper.

## 2. Methods

### 2.1. Model Architecture

The model shown in Figure 1 integrates three convolutional neural network feature extraction modules, a recursive feature elimination module based on feature weights (FW-RFE), and a fully connected layer. Three convolutional neural networks are trained independently without interfering with each other. The three fully convolutional neural networks are prepared separately from each other in such a way that the spatially fused information features of each convolutional neural network are different, so the output of each part of the features expresses a different meaning. This module analyzes the characteristics of the weights to select the most relevant features while eliminating unrelated and redundant features by implementing adaptive features filtering. Finally, the three retained feature vectors are concatenated and output to the fully connected layer to obtain the final classification result.

### 2.2. Extraction of Spatial Data: CNN-Based Feature Extraction

CNNs [25] are one of the most widely used artificial neural networks. Convolutional neural networks have two types of layers: convolutional and pooling. The convolutional layer has a number of filters via which the neurons link the input data points directly and multiply them by the weights. The convolutional layer is calculated as follows: *y* is the convolutional layer’s output, *n* is the number of input samples, *x${}_{i}$* is the convolutional layer’s ith input number, *w${}_{i}$* is the weight corresponding to that number, *f* is the activation function, and *b${}_{i}$* is the offset corresponding to that number.    
(1)$$y=f(\sum_{i=1}^{n}w_{i}*x_{i}+b_{i})$$

The pooling layer frequently follows the convolution layer. To achieve the goal of lowering the matrix size while holding the matrix depth constant, adjacent pooling units collect inputs via repeated row or column panning. The maximum pooling function [26] is determined as follows, where *P${}_{i}$* is the output of the ith feature of the pooling layer, *y${}_{i}$* is the ith feature input of the pooling layer, and *S* is the pooling block size.
(2)$$P_{i}=max\left(y_{i}\right),y_{i}\in S$$

All neurons in the filter of the convolutional layer share weights, enabling the sharing of channel information and facilitating the spatial fusion of information from different ECG leads. The output features become increasingly biased to express high-level characteristics as the number of convolutional layers grows, while the expression of low-level features is repressed. In earlier studies, low-level feature extraction was always ignored [27,28], and their performance proved the importance of low-level features. Low-level features contain more initial information about the data, but are noisier due to less convolution. High-level features include more contextual information [29], with abstract information more closely resembling human understanding. Therefore, different convolutional neural networks have various features for spatially fusing information, and different convolutional neural networks have different meanings expressed by the output features. Furthermore, various convolutional neural networks produce features with different degrees of noise, and these insignificant features impact the model’s generalization ability.

### 2.3. Feature Optimization: Recursive Feature Elimination Based on Feature Weights

The machine learning feature extraction method [30,31] is to process according to the parameters given by machine learning to filter the features.This paper chooses to perform feature optimization to improve the quality of features and select the most relevant features while eliminating unrelated ones. This paper proposes a recursive feature elimination method based on feature weights, called FW-RFE. The method analyzes the weight values of the features obtained from the CNN model to obtain the feature ranking, whereby the lowest-ranked features are selected and removed. The feature set is fed into the fully connected layer for classification at each iteration. The method identifies the optimal set of features extracted from the CNN model based on the classification results. FW-RFE achieves the adaptive filtering of features, and the features retained are more representative and effective.The detailed algorithm of FW-RFE is shown in Algorithm  1.
**Algorithm 1** FW-RFE**Input:** The set of output features of the pooling layer and sample labels,
$${\left\{(P_{i},y_{i})\right\}}_{\left(i=1\right)}^{Num\_l},P_{i}\in {\left\{p_{1},p_{2},\cdots ,p_{j}\right\}}_{j=1}^{Num\_f},y_{i}\in \left\{+1,0\right\}$$
   and the set of convolutional layer feature weights,
$${\left\{\left[{w_{1}}^{1},{w_{2}}^{1},\cdots {w_{10}}^{1}\right],\left[{w_{1}}^{2},{w_{2}}^{2},\cdots {w_{10}}^{2}\right],\cdots ,\left[{w_{1}}^{j},{w_{2}}^{j},\cdots {w_{10}}^{j}\right]\right\}}_{\left(j=1\right)}^{Num\_f}$$
**Output:** Feature Sorted Set R;
1:Initial feature set S
$$S={\left\{s_{i}\right\}}_{i}^{num\_l},s_{i}=\left\{p_{1}^{i},{p_{2}}^{i},\cdots ,{p_{j}}^{i}\right\}$$
and accuracy set V=⌀;2:**while**(S≠ϕ)**do**3:    Calculate the ranking criterion score:$c_{i}=\sum_{n=1}^{10}{\left(w_{n}^{j}\right)}^{2}$;4:    Find the feature with the smallest ranking score:
h=minC5:    Extract features $s_{i}=s_{i}-p_{h}^{i}$, updata S;6:    Take S as the input of the fully connected layer, use sigmoid regression function to achieve binary classification: $sigmoid\left(S_{i}\right)=\frac{1}{1+e^{-S_{i}}}$7:    Update accuracy set: $V=\left[acc,V\right]$8:**end while**9:When the accuracy rate in the accuracy set V is the highest, the feature set in S is selected as the output result


FW-RFE can assist the model in focusing on the signal’s key information, hence boosting detection performance.

## 3. Experiment

### 3.1. ECG Collection

The ECG heartbeat signals are obtained from PTB-XL. The PTB-XL ECG dataset is a large dataset of 21,837 clinical 12-lead ECGs from 18,885 patients. These recordings were interpreted and validated by up to two cardiologists. We considered two main paradigms—DA and DB—to evaluate the proposed model.

With a total of 17,259 pieces of 10-second ECG data, this research uses Śmigiel S’s data selection approach [32] in the DA paradigm and picks ECG data with 100% confirmation by clinicians that the illness conclusion is true. Table 1 shows the diagnostic distribution of the chosen data. The datasets are separated into two categories: normal datasets (7185 items) and aberrant datasets (6925 items are randomly selected). The sequence of the data sets is scrambled, and 5198 abnormal 12-lead ECG data and 5198 normal ECG data are used to create a training set, 577 abnormal 12-lead ECG data and 577 normal ECG data are used to create a validation set, and 1150 abnormal ECG data and 1410 normal ECG data are used to generate the test set. The recordings were sampled at a frequency of 500 Hz.

Simultaneously, when different samples of the same patient occur in the training set, validation set, and test set simultaneously, the model’s generalization performance may be reduced. To eliminate the impact of this one factor, we evaluated the model using the DB paradigm.

Only one ECG data was chosen for each patient in the DB paradigm, with 100 percent physician validation of the specific ECG data for the illness findings, totaling 14,908 10-s ECG data. Table 2 shows the diagnostic distribution of the chosen data. The datasets are separated into two categories: normal datasets (6590 items) and aberrant datasets (6590 items are randomly selected). The sequence of the data sets is scrambled, and 5031 abnormal 12-lead ECG data and 5031 normal ECG data are used to create a training set, 559 abnormal 12-lead ECG data and 559 normal ECG data are used to create a validation set, and 1000 abnormal ECG data and 1000 normal ECG data are used to generate the test set. The recordings were sampled at a frequency of 500 Hz.

Normal ECGs were labeled as 0, and abnormal ECGs were labeled as 1. The data from each lead of the ECG was also filtered using a band-pass filter to achieve noise reduction of the ECG signal and improve the signal-to-noise ratio. Figure 2 shows a selection of filtered samples from the PTB-XL database of five ECG second lead recordings (i.e., normal and other abnormal cases).

### 3.2. Model Setting

Figure 3 depicts the model parameters of the three convolutional neural networks: CNN-A, CNN-B, and CNN-C. A LeakyReLU [33,34] follows each convolutional layer. A fully connected layer is the ultimate level in all neural network frameworks. This study employs two hidden layers. The first hidden layer uses Relu and the second hidden layer uses a sigmoid regression function to achieve binary classification. The extracted features are the output of the ECG signal after it has passed through the last pooling layer of each model.

In the FW-RFE part, the training set and the validation set are input for training and validation after a three-part feature set obtained from CNN-A, CNN-B, and CNN-C. The fully connected layer is a hidden layer with a sigmoid regression function to implement binary classification. The last feature in the ranking is eliminated in each recursive elimination iteration, making the features adaptively filtered to obtain the best features in the feature set extracted by each CNN model.

In the diagnostic part, a hidden layer consisting of a sigmoid regression function implementing a binary classification is used.

The Adam optimizer [35] was used to train the whole model, with the beginning learning rate for the diagnostic module set to 0.0001 and 0.001 for the rest of the model. The Adam optimizer was applied to balance gradient updates across categories, thus mitigating the detrimental effects of data imbalance. Furthermore, the learning rate of the model was multiplied by 0.1 without boosting the accuracy of the validation set after every five epochs. The L2 loss of all parameters in the model was multiplied by a factor of 0.001 to prevent certain parameters from dominating the computation by being too large. A binary cross-entropy loss function was also used as the loss function throughout the model. EarlyStopping was used throughout the model, using the validation set accuracy as a reference, and training was stopped if there was no growth in 15 iterations. This prevents the network from overfitting.

## 4. Results

### 4.1. Evaluation Indicators

In this section, four evaluation indicators are investigated to evaluate the performance of the presented models in the following sections. The four evaluation indicators are Subset Accuracy, Precision, Recall and F1 score, which are introduced as follows.
(3)$$SubsetAccuracy=\frac{TP+TN}{TP+TN+FP+FN}$$
(4)$$Precision=\frac{TP}{TP+FP}$$
(5)$$Recall=\frac{TP}{TP+FN}$$
(6)$$F1=\frac{2*Precoision*Recall}{Precoision+Recall}$$

TP (Ture Positive) represents the number of positive samples correctly diagnosed by the model; TN (Ture Negative) represents the number of negative samples correctly diagnosed by the model; FP (False Positive) represents the number of positive samples incorrectly diagnosed by the model; FN (False Negative) represents the number of negative samples incorrectly diagnosed by the model.

### 4.2. Feature Elimination Performance

Figure 4 shows the process of feature adaptive screening. The number of removed features is demonstrated by the horizontal coordinate, while the vertical coordinate indicates the accuracy of the feature set on the validation set. For the first 250-or-so features, CNN-A’s accuracy rate hovers around 1% before plummeting in a sudden downward trend. The accuracy of CNN-B and CNN-C varies between 1% and 1% for the first 160 and 350 features, respectively, before showing a cliff-down trend.

Finally, we obtain the features preserved by the model with the best accuracy on the validation set. The DA paradigm retains 125 features with CNN-A, 89 features with CNN-B, and 223 features with CNN-C. Through CNN-A, the DB paradigm retains 346 properties. CNN-B retains 116 features, but CNN-C retains 229 features.

### 4.3. Classification Performance

Table 3 shows the differences between the proposed model and the traditional CNN model in terms of Subset Accuracy, Precision, Recall, and F1 score in the DA paradigm. The suggested model increases Subset Accuracy by 0.58%, Precision by 0.7%, Recall by 0.3%, and F1 score by 0.5%, as shown in Table 3.

Table 4 shows the differences between the proposed model and the traditional CNN model in terms of Subset Accuracy, Precision, Recall, and F1 score in the DB paradigm. Table 4 shows that the suggested model increases Subset Accuracy by 2.3%, Precision by 2.3%, Recall by 1.3%, and F1 score by 2.3%.

In all respects, the findings demonstrate that the model presented in this study outperforms existing reference CNN models.

### 4.4. Model Test on PTB-XL

Table 5 compares the diagnostic ability of the five reference models and CNN-FWS for abnormal and normal ECG, based on PTB-XL. The results are expressed using Recall and F1 scores, which represent the diagnostic ability for abnormal and normal ECG data. Figure 5 shows the confusion matrix for evaluating the network results.

In the DA scenario, the findings reveal that CNN-FWS beats the other reference models in Recall and F1 scores. Recall is 18.4% higher, and F1 score is 14.9% higher when compared to DLECG-CVD [36]. Recall is 20% better, and F1 score is 16.4% higher when compared to CIGRU-ELM [37]. Compared to IBECG-SP [38], Recall is 19.5% higher, and F1 score is 15.9% higher when both employ fully connected layers for classification diagnosis. This set of comparisons means that the model presented in this study is better at obtaining features using CNNs. In addition, while extracting features using CNN, F1 score is 7.5% higher than MLBF-Net [39]. Recall increases by 2.5%, and F1 score is 0.3% better when compared to CNN with entropy features [32]. The performance of CNN-FWS on Recall and F1 scores did not degrade much in the DB paradigm.

## 5. Discussion

When different samples of the same patient appear in the training set, validation set, and test set simultaneously, it may suppress results for the generalization performance of the model. To exclude this effect, we introduced the DB paradigm dataset based on the DA paradigm dataset.

From Table 3 and Table 4 it is easy to see that different CNNs have different diagnostic capabilities for normal and abnormal ECGs with the same dataset. This result indicates differences in the features extracted by different CNNs. Therefore, concatenating the features extracted by different CNNs can expand the content of the features. By concatenation, it is possible to increase the generalization ability of the model. However, the presence of varying degrees of noise features in the features extracted by convolutional neural networks tends to affect the diagnostic capability of the model.

As shown in Figure 4, with each recursive elimination, the feature set extracted by the convolutional neural network fluctuates up and down for ECG normal and abnormal diagnosis accuracy. By retaining the feature set with the highest accuracy for normal and abnormal ECG diagnosis, we select the most relevant features while eliminating unrelated and redundant features, further optimizing the information and attaining adaptive filtering of features. The final features that are left are more representative and effective. The results in Table 3 and Table 4 also demonstrate that CNN-FWS is a better model for ECG normal and abnormal diagnosis than CNN.The FW-RFE in this paper can directly target the adaptive screening of features by extracting parameters from the CNN without the intervention of machine learning methods, reducing the complexity of the ECG positive anomaly classification algorithm computation and memory.

As the results in Table 5 show, CNN-FWS also demonstrates good advantages for normal and abnormal ECG diagnosis by compared with other models. Compared with the literature [36,37,38], both the Recall and F1 scores are greater than 14%, reflecting that the CNN in this paper is a better way of obtaining features. From Figure 5, we can find that with the use of CNN to extract features, this paper effectively improves diagnostic efficiency of ECG abnormalities without sacrificing diagnostic efficiency for normal ECG. Therefore, the method in this paper effectively reduces the leakage of abnormal ECG, which will significantly improve the efficiency of ECG diagnosis for physicians. Furthermore, comparing the results under the DA and DB paradigms, the Recall and F1 scores of CNN-FWS did not drop substantially when the amount of data decreased and when the possible effects of the same patient data on the model were excluded. This result indicates that CNN-FWS is stable and has excellent generalization performance for ECG diagnosis.

As a result, CNN-FWS increases the information validity of features in general by using CNN feature extraction and FW-RFE to select the most relevant features while eliminating unrelated and redundant features and achieving adaptive filtering of features, resulting in a model with superior generalization performance than previous methods. At the same time, this method reduces the complexity of the CNN feature screening algorithm computation and memory, effectively reducing the underdiagnosis of abnormal ECG, which is conducive to significantly reducing the difficulty of physicians’ diagnosis.

## 6. Conclusions

In this study, CNN-FWS combining a convolutional neural network (CNN) and recursive feature elimination based on feature weights (FW-RFE) is used for ECG normal and abnormal classification. A large amount of feature information is obtained through CNN’s excellent feature extraction properties. Features are adaptively filtered and select the most relevant features while eliminating unrelated and redundant features through FW-RFE. These modules with different functions are integrated into a single neural network architecture, resulting in superior classification performance, especially in detecting normal and abnormal ECG. In terms of normal and abnormal ECG classification, the F1 score was 0.902, and Recall was 0.889. In summary, the presented method provides an excellent solution to the problem of detecting normal and abnormal ECGs as an entirely data-driven end-to-end method that requires no human intervention. It provides examples for other signal processing problems involving the spatial fusion of signals.

## Figures and Tables

**Figure 1 entropy-24-00471-f001:**
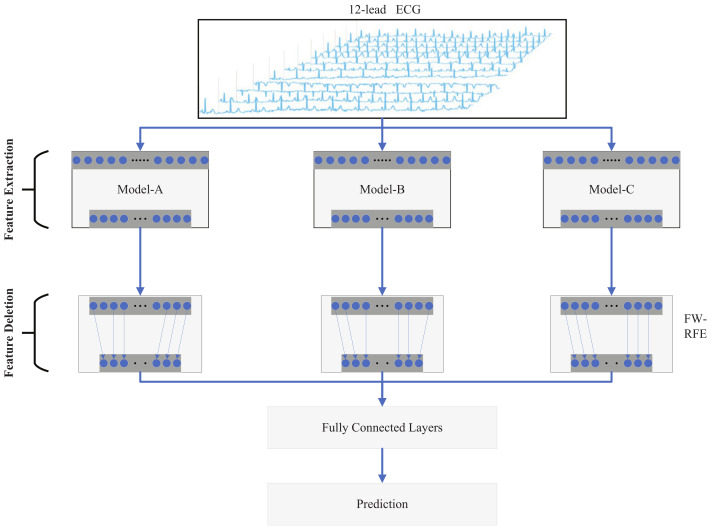
Architecture for the proposed model. The 12-lead ECG is fed into a convolutional layer to obtain different features. The feature elimination module accepts the output of the convolutional layer and retains the most representative features. The final classification result is output through the fully connected layer.

**Figure 2 entropy-24-00471-f002:**
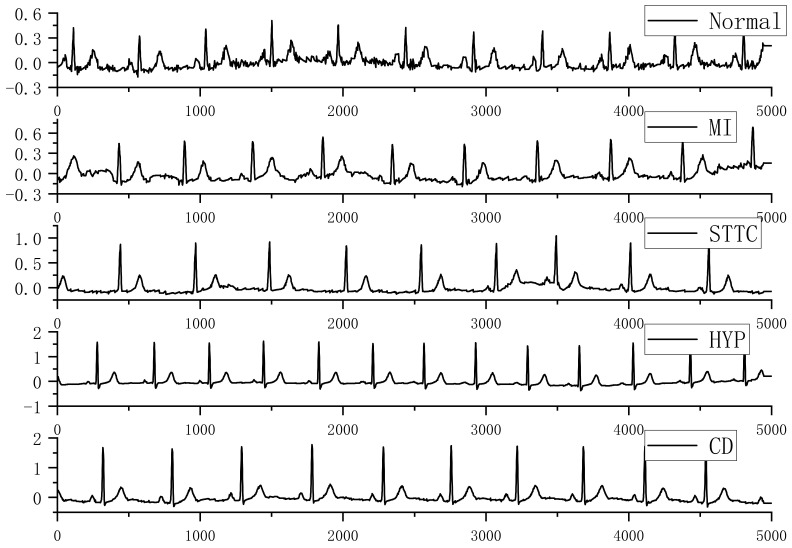
Normal and abnormal class filtered samples from the PTB-XL dataset.

**Figure 3 entropy-24-00471-f003:**
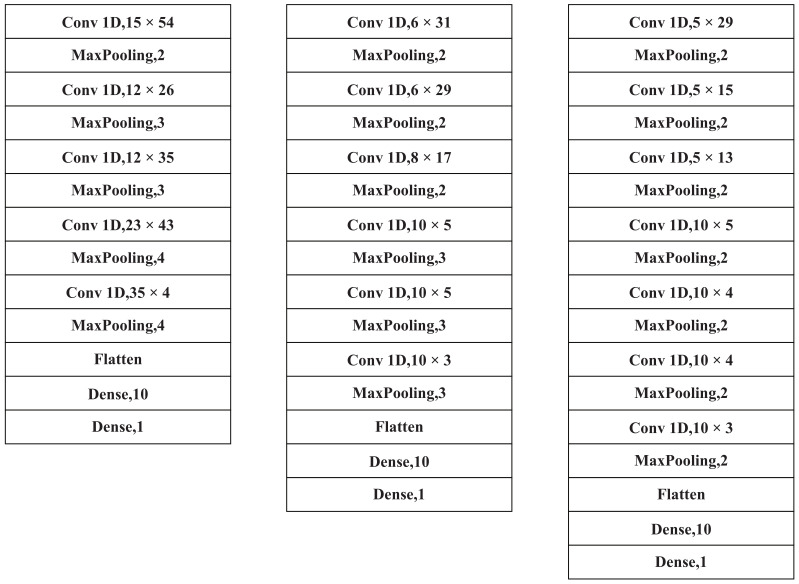
Convolutional neural network layer configurations. CNN-A, CNN-B, and CNN-C designs are shown from left to right. A convolutional layer with 54 kernels of size 15 is designated as “Conv1D,1554”. A maximum pooling layer with a step size of 2 is called “Maxpooling”. A fully linked layer is referred to as “Faletten”. “Dense,10” refers to a buried layer with a 10 output.

**Figure 4 entropy-24-00471-f004:**
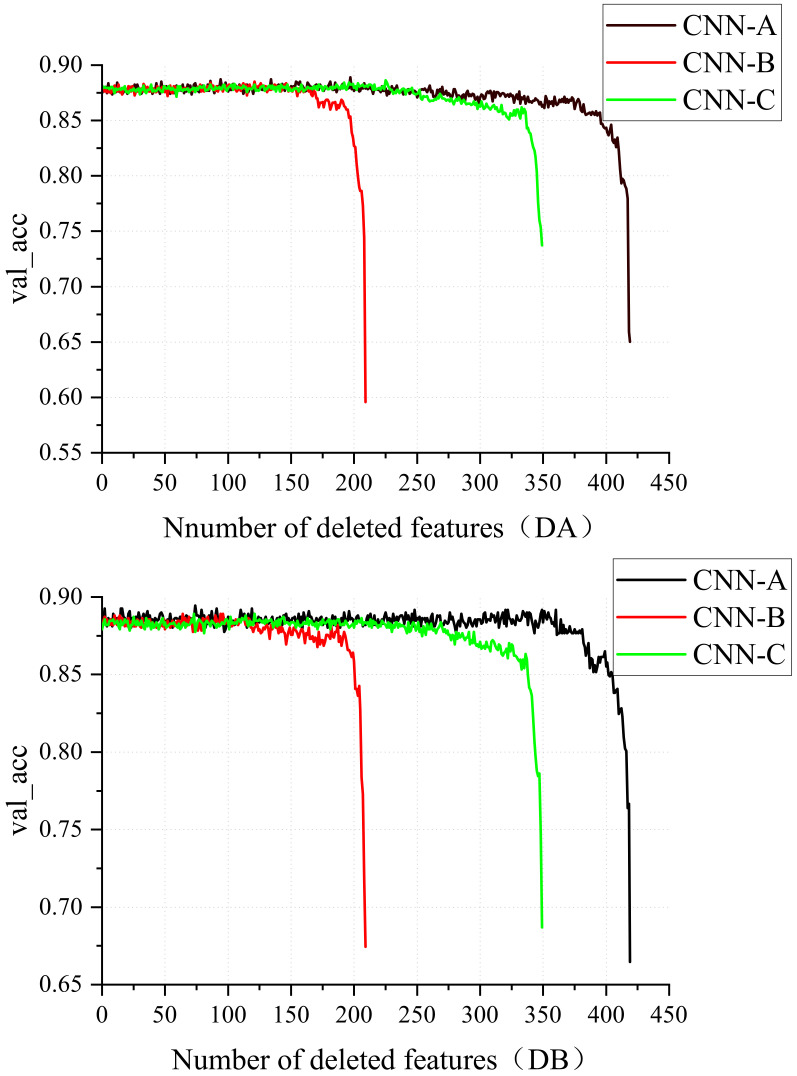
The process of removing features. The number of removed features is shown by the horizontal coordinate, while the vertical coordinate indicates the accuracy of the feature set on the validation set.

**Figure 5 entropy-24-00471-f005:**
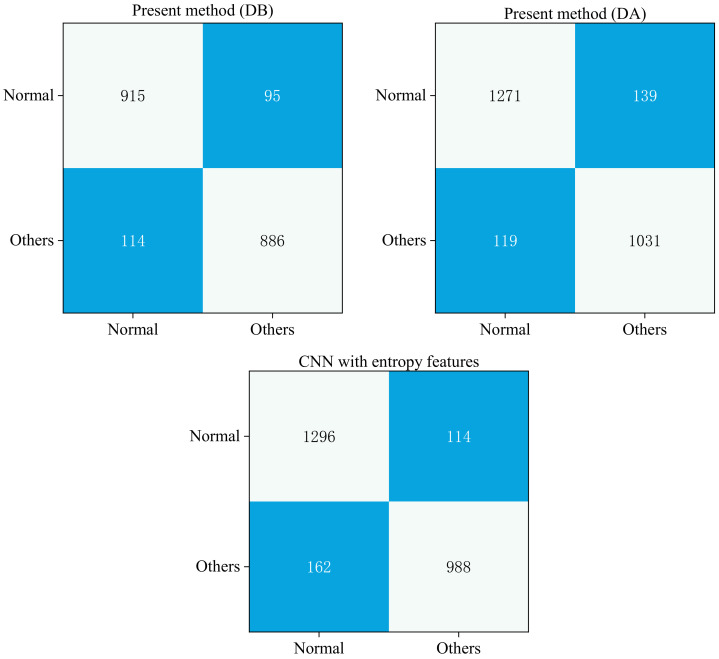
Confusion matrix for evaluating network results.

**Table 1 entropy-24-00471-t001:** A summary table with the breakdown of the five classes of DA.

Records	Description
7185	Normal ECG (Normal)
2984	Myocardial Infarction (MI)
4579	ST/T Change (STTC)
4787	Conduction Disturbance (CD)
1473	Hypertrophy (HYP)

**Table 2 entropy-24-00471-t002:** A summary table with the breakdown of the five classes of DB.

Records	Description
6590	Normal ECG (Normal)
2363	Myocardial Infarction (MI)
3735	ST/T Change (STTC)
4014	Conduction Disturbance (CD)
1201	Hypertrophy (HYP)

**Table 3 entropy-24-00471-t003:** The performance of different models used in this paper on the test set (DA).

Model	Accuracy	Precision	Recall	F1 Score
Present method	89.92%	0.901	0.914	0.907
CNN-A	88.25%	0.887	0.876	0.878
CNN-B	89.34%	0.894	0.911	0.902
CNN-C	87.58%	0.871	0.900	0.885

**Table 4 entropy-24-00471-t004:** The performance of different models used in this paper on the test set (DB).

Model	Accuracy	Precision	Recall	F1 Score
Present method	90.05%	0.915	0.889	0.902
CNN-A	87.75%	0.879	0.876	0.878
CNN-B	87.40%	0.882	0.868	0.875
CNN-C	87.75%	0.892	0.867	0.879

**Table 5 entropy-24-00471-t005:** The performance of different models on the test set.

Approach	Recall	F1 Score
Present method (DB)	0.889	0.902
Present method (DA)	0.914	0.907
IBECG-SP [38]	0.719	0.748
DLECG-CVD [36]	0.730	0.758
MLBF-Net [39]	0.714	0.832
CIGRU-ELM [37]	-	0.743
CNN with entropy features [32]	0.889	0.904

## Data Availability

The data presented in this study are available upon request of the corresponding author or on the website The data presented in this study are available upon request from the corresponding author (https://www.physionet.org/content/ptb-xl/1.0.1/ (accessed on 2 March 2022)).

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
