# Peer review of "CNN-FWS: A Model for the Diagnosis of Normal and Abnormal ECG with Feature Adaptive"

_entropy, 2022, doi:10.3390/e24040471_

Round 1

Reviewer 1 Report

The paper is very interesting and it adopts the CNN and RNN for feature extraction and elimination for diagnosis of disease. However, I suggest the authors to provide more explanation why they select the initial parameters in NN for training and validation. For example, they selected around 14100 data for training the NN but they collected more than 17000 data from patients. Please explain more why some data are filtered by what criteria that the authors set. Overall, the paper is the other example for demonstrating the capabilities of CNN and RNN in image processing.

Author Response

Dear Reviewers,

Thank you for the reviewers’ comments concerning our manuscript entitled “CNN-FWS: A model for the diagnosis of normal and abnormal ECG with feature adaptive”. (Manuscript ID: entropy-1647653).

These comments are all valuable and very helpful for revising and improving our paper, as well as the important guiding significance to our researches. We have studied comments carefully and have made corrections accordingly which we hope meet with your approval. The responds to the reviewer’s comments are as follows:

In paragraph 3 of the article 2.1. ECG collection, we explain the reason we selected around 14100 data for training the NN but they collected more than 17000 data from patients. Perhaps, this paragraph is inconspicuously placed and under-emphasized. Therefore, we have modified and added a paragraph, which is the first paragraph of the Discussion in Chapter 4.

The added statement is as follows:

When different samples of the same patient may appear in the training set, validation set, and test set simultaneously, it may bring suppressed results for the generalization performance of the model. To exclude the effect of this one reason, we introduced the DB paradigm dataset based on the DA paradigm dataset.

Once again, thank you very much for your comments and suggestions.

Kind regards,

Mr. Lv

Reviewer 2 Report

General summary. Publication is possible after revision.

Comments for authors.  The results of this CNN-FWS study are of undoubted interest, since they have a high practical value. In this study, CNN-FWS combining a convolutional neural network and recursive feature elimination based on feature weights provides a solution to the problem of fast and accurate detection for ECG normal and abnormal classification without physician intervention. The reliability of the obtained model is confirmed by the large dataset of 21837 clinical 12-lead ECGs from 18885 patients, by the methods of mathematical data processing, by the results of checking the diagnostic accuracy of the model.

Remarks. The Discussion is very briefly, there is not enough data about works of other authors. There is no analysis of advantages/disadvantages of your model in comparison with models of other authors.

Author Response

Dear Reviewers,

Thank you for the reviewers’ comments concerning our manuscript entitled “CNN-FWS: A model for the diagnosis of normal and abnormal ECG with feature adaptive”. (Manuscript ID: entropy-1647653).

These comments are all valuable and very helpful for revising and improving our paper, as well as the important guiding significance to our researches. We have studied comments carefully and have made corrections accordingly which we hope meet with your approval. The responds to the reviewer’s comments are as follows:

In order to fully compare the implementation data, we try to add the following statement in Chapter 4 Discussion:

Compared with machine learning feature extraction methods [35] [36], the FW-RFE taken only in this paper can directly target the adaptive screening of features by extracting parameters from the CNN without the intervention of machine learning methods, reducing the complexity of the ECG positive anomaly classification algorithm computation and memory. (At the end of the third paragraph)

Compared with the literature [30, 31, 33], both Recall and F1 scores are greater than 14%, reflecting that the CNN taken in this paper is a more excellent way to obtain features. From Figure 5, we can find that with the use of CNN to extract features, this paper effectively improves the diagnostic efficiency of ECG abnormalities without sacrificing the diagnostic efficiency for normal ECG. Therefore, the method in this paper effectively reduces the leakage of abnormal ECG, which will significantly improve the efficiency of ECG diagnosis for physicians. (Beginning of the second sentence of paragraph 4)

At the same time, this method reduces the complexity of the CNN feature screening algorithm computation and memory, effectively reducing the underdiagnosis of abnormal ECG, which is conducive to significantly reducing the difficulty of physicians' diagnosis. (At the end of the last paragraph)

Through the above discussion, the advantages of our model are further demonstrated compared to other authors' models.

Once again, thank you very much for your comments and suggestions.

Kind regards,

Mr. Lv
